# FD-2, an Anticervical Stenosis Device for Patients Undergoing Radical Trachelectomy or Cervical Conization

**DOI:** 10.3390/bioengineering10091032

**Published:** 2023-09-01

**Authors:** Seiji Mabuchi, Shoji Kamiura, Takuya Saito, Hayato Furukawa, Azusa Abe, Takashi Sasagawa

**Affiliations:** 1Department of Gynecology, Osaka International Cancer Institute, 3-1-69, Otemae, Chuo-ku, Osaka-shi 541-8567, Japan; kamiura-sh@oici.jp; 2Medical Technical Sec., Fuji Latex Co., Ltd., 1705 Chizuka-machi, Tochigi 328-0135, Japan; t-saitou@fujilatex.jp (T.S.); ha-furukawa@fujilatex.jp (H.F.); 3Quality Control Sec., Fuji Latex Co., Ltd., 1705 Chizuka-machi, Tochigi 328-0135, Japan; a-tanifuji@fujilatex.jp; 4Fuji Latex Co., Ltd., 3-19-1 Kanda Nishiki-cho, Chiyoda-ku, Tokyo 101-0054, Japan; sasagawa@fujilatex.jp

**Keywords:** cervical stenosis, radical trachelectomy, conization, cervical cancer, antistenosis device

## Abstract

This study aimed to introduce FD-2, a newly developed anticervical stenosis device for patients with cervical cancer undergoing radical trachelectomy. Using ethylene-vinyl acetate copolymers, we developed FD-2 to prevent uterine cervical stenosis after radical trachelectomy. The tensile test and extractables and leachables testing were performed to evaluate FD-2’s safety as a medical device. FD-2 was indwelled in three patients with cervical cancer during radical trachelectomy and its utility was preliminarily evaluated. FD-2 consists of a head (fish-born-like structure), neck (connecting bridges), and body (tubular structure); the head is identical to FD-1, an intrauterine contraceptive device. FD-2 passed the tensile test and extractables and leachables testing. The average time required for the application or removal of FD-2 in cervical cancer patients was less than 10 s. The median duration of FD-2 indwelling was 8 weeks. No complications, including abdominal pain, pelvic infections, or hemorrhages, associated with FD-2 indwelling were reported. At the 3–12-month follow-up after the radical trachelectomy, no patients developed cervical stenosis or experienced dysmenorrhea. In conclusion, we developed FD-2, a novel device that can be used for preventing cervical stenosis after radical trachelectomy for uterine cervical cancer.

## 1. Introduction

Cervical stenosis is a common and troublesome clinical problem that can develop after radical trachelectomy for cervical cancer or conization for cervical intraepithelial neoplasia (CIN). A recent report suggested that cervical stenosis is caused by the excessive wound-healing response and cervical canal adhesions [1]. The reported incidence of cervical stenosis after radical trachelectomy or cervical conization differs depending on the definition employed, ranging from 0% to 73.3% [2,3] or 4% to 17% [4,5], respectively.

Although its severity varies from patient to patient, in premenopausal women, cervical stenosis may give rise to symptoms such as amenorrhea, dysmenorrhea, and hematometra, which compromise the quality of life and reproductive functions, as they seriously impede embryo transfers or intrauterine insemination [2,3,4,5]. Moreover, it can hinder gynecological evaluations, such as cervical cytology, colposcopic examination, endometrial biopsy, and hysteroscopy. When it develops, cervical stenosis requires dilation of the cervical ostium; it must be performed several times for an optimal result in some cases [2,3,4,5].

Preventing cervical stenosis development is evidently the most effective approach. Although efforts to prevent stenosis have been made using different types of antistenosis devices [6,7,8,9,10,11,12], a standard method of prevention has yet to be established. An ideal antistenosis device should satisfy the following requirements: sustainably dilate the resected cervical canal, allow menstrual blood to drain, and be indwelled in the uterus for a long time.

Recently, we developed an anticervical stenosis device, FD-2, which is composed of inert elastic ethylene-vinyl acetate (EVA) copolymers, a soft material that can be safely used in biomedical applications [13]. In the current study, we show its design, functionality, durability, and its potential usability in patients.

## 2. Materials and Methods

### 2.1. FD-2, a New Anticervical Stenosis Device

FD-2, an indwelling intrauterine tube, was jointly developed by Dr. Seiji Mabuchi and Fuji Latex Co., Ltd. (Tokyo, Japan) in 2022 to prevent cervical stenosis development after radical trachelectomy or cervical conization (trademark registration: 2022-120518). A flow diagram of the FD-2 development is shown in Appendix A. Although its prototype, FD-1, had been approved as an intrauterine contraceptive device (IUD) in Japan [14], FD-2 has not been approved as an anticervical stenosis device but is currently being considered for a clinical trial in cervical cancer patients undergoing radical tracherectomy or CIN patients undergoing cervical conization.

FD-2 is composed of inert elastic ethylene-vinyl acetate (EVA) copolymers, a soft material that can be safely used in biomedical applications, including contraception [14]. As shown in Figure 1, FD-2 consists of three parts: a body, neck, and head. The head, a fish-born-like structure, is identical to FD-1, an IUD that was developed more than 50 years ago using EVA copolymers and has been widely used in Japan [14]. Since the pliable arms on the side are designed to bend back and forth, as observed during FD-1 insertion, FD-2 insertion into the uterine cavity ought to be easy by holding the body of FD-2 and pushing its head into the uterine cavity through the cervical canal. FD-2 removal ought to be easy as well by gently pulling the body using forceps. The pliability of these arms can be achieved by the highly elastic nature of EVA copolymers. In a physiological condition without any forces on the FD-2, the arms do not bend and prevent FD-2 from falling out and help maintain the body part in the proper position within the cervical canal.

### 2.2. Tensile Properties

Uniaxial tensile tests were conducted using TechnoGraph TGI-1kN (MinebeaMitsumi Inc., Tokyo, Japan). The sample was conditioned in a thermostatic bath (36.5 °C) for 72 h before initiating the test. Subsequently, the tensile force was applied in the longitudinal axis to elongate FD-2, a preload of 1.0 newton (N) was applied, and the tests were run at a crossbar rate of 200 mm/min. Based on the guideline from the Pharmaceutical Affairs Bureau in Japan (notification no. 1228 promulgated in 1974), a breaking load greater than 9.8 N is considered passing.

### 2.3. Extractables and Leachables Testing

Extractables and leachables testing was conducted based on the guideline of the Pharmaceutical Affairs Bureau in Japan (notification no. 1228 promulgated in 1974) as well as that from the Ministry of Health, Labour and Welfare of Japan (promulgated in 1971). Ten samples (FD-2) were stored in 100 mL of water for 72 h at 50 °C before initiating the test (test solution). A blank solution containing water alone was prepared in the same manner (reference solution). After the solutions were kept at room temperature, the following tests were conducted: transparency, pH, foaming, heavy metal, potassium permanganate-reducing substance, and residue after evaporation tests.

### 2.4. Patients

FD-2 was used during radical trachelectomy with or without pelvic lymphadenectomy in three patients with early-stage cervical cancer. The radicality of the procedure was classified according to the Querleu–Morrow classification [15]. Radical trachelectomy was performed via vaginal or retroperitoneal approaches using the same procedures that were reported previously [16,17]. Written informed consent for the use of FD-2 during radical trachelectomy and the publication of a case series and accompanying images were obtained from all patients. The FD-2 was removed after patients experienced two menstrual cycles. In patients who experienced two menstrual cycles within a short period, FD-2 remained indwelled for at least 1 month.

### 2.5. Definition of Cervical Stenosis

Cervical stenosis was defined as follows: (1) the patient complained of abnormal menstruation, including dysmenorrhea or amenorrhea, and required cervical dilation for treatment or (2) the cervical canal did not permit the insertion of a cell brush or cervical dilator. When one of the two abovementioned definitions was met, cervical stenosis was diagnosed.

## 3. Results

### 3.1. Manufacturing FD-2

FD-2, comprising a body, neck, and head composed of EVA copolymers, can be used to prevent uterine cervical stenosis after radical trachelectomy (Figure 1A,B). The body has a tubular structure and maintains cervical ostium dilation, inhibits cervical canal adhesion, and prevents cervical stenosis development. Its length is 46.5 mm, with inner and outer diameters of 3 mm and 5 mm, respectively. The neck consists of two bridges that connect the head and body. A pair of triangular windows between the bridges allow for the drainage of menstrual blood into the vagina (Figure 1A,B). The head, which is inserted into the uterine cavity (Figure 1C), helps maintain the body in the proper position within the cervical canal.

### 3.2. Bench Testing

Bench testing was performed to evaluate the durability and safety of FD-2 as a medical device. Uniaxial tensile tests were conducted using five FD-2 devices to evaluate their tensile property (Figure 2). The breaking load of the five FD-2 devices ranged from 35.1 to 41.8 N, which was significantly higher than the reference score. Subsequently, we performed extractables and leachables testing. As shown (Table 1), FD-2 met all the requirements in the testing. Collectively, the results obtained from bench testing (Table 1 and Figure 2) show that FD-2 meets all the criteria required for a medical device and indicate the potential safety and durability of FD-2 as an anticervical stenosis device.

### 3.3. Case Series

Between February 2022 and November 2022, three women (aged 32–36) with FIGO stage IA2 (n = 1) or IB1 (*n* = 2) cervical cancer who desired to preserve their fertility underwent a radical trachelectomy with or without pelvic lymphadenectomy via an extraperitoneal (*n* = 2) or a vaginal (*n* = 1) approach. In all the cases, the cervix was amputated at least 10 mm below the isthmus. After transection of the uterine cervix, FD-2 was inserted into the uterine cavity (Figure 1C and Figure 3), where it remained for 5–10 weeks. The time required for the insertion or removal of FD-2 was less than 10 s in all patients. No intraoperative or postoperative complications occurred. During the indwelling period, all patients experienced two menstrual cycles and did not report any menstrual problems. In addition, other complications including abdominal discomfort, abdominal pain, pelvic infections, and hemorrhages associated with FD-2 indwelling were not reported. At the follow-up, 9–18 months after the radical trachelectomy, no patients developed cervical stenosis or experienced dysmenorrhea.

## 4. Discussion

Although radical trachelectomy is a safe alternative to radical hysterectomy in young patients with cervical cancer who wish to preserve fertility, specific problems may be encountered after this procedure. Cervical stenosis is one of the most common and important complications. It is accompanied by surgical scarring, contracture, and regeneration of the vaginal epithelium covering the new internal os, particularly where the cervix and vagina are reconstructed [18]. According to a meta-analysis by Li et al. [2], 10.5% of patients who undergo radical trachelectomy develop cervical stenosis. The incidences of cervical stenosis after abdominal radical tracherectomy, laparoscopic radical trachelectomy, and vaginal radical trachelectomy are 11.0%, 9.3%, and 8.1%, respectively. Cervical stenosis can also develop in patients after cervical conization, with a reported global incidence of 7.1% [19].

Although the severity of cervical stenosis varies from patient to patient, in cases of severe cervical stenosis, dilatation of the cervical ostium must be performed, followed by the application of indwelling antistenosis tools. To date, in the absence of sufficient clinical evidence, different types of antistenosis tools have been used to prevent restenosis after dilation: a Foley catheter [6], the levonorgestrel-releasing intrauterine system [7], IUDs [8,9], a self-expanding nitinol stent [10], a Smit sleeve [11], and a Petit-Le Four pessary [12]. A systematic review of published retrospective studies suggested that in patients with cervical cancer after radical trachelectomy, the use of the antistenosis tools could effectively reduce the occurrence of cervical stenosis (4.6% in patients with antistenosis tools versus 12.7% in those without antistenosis tools, *p* < 0.01), indicating the possibility that mechanical cervical dilatation is effective in preventing cervical stenosis development [2]. Globally, the most frequently used antistenosis tool in patients after radical trachelectomy is the Foley catheter, followed by an IUD with or without a catheter, and a Smit sleeve [2]. In Japan, although nationwide surveillance has not been conducted, in a relatively large study including 65 cervical cancer patients who underwent radical tracherectomy, the Forley catheter (10–12 Fr), to prevent cervical stenosis, and the intrauterine contraceptive device (FD-1; Fuji Latex Co., Tokyo, Japan), to prevent intracavity adhesion, were routinely utilized [3]. However, these devices have limitations. They have been used for unapproved indications (off-label usages) and, thus, their safety in an oncological setting remains unknown; the Foley catheter can easily fall out upon ambulation, even in cases where the Foley catheter is sutured to the uterine cervix; the nylon threads tied with IUDs are too thin to maintain the dilatation of the cervical ostium, and some IUDs are difficult to remove [20]; and the use of a Smit sleeve is complex and expensive. This highlights the need for new antistenosis devices that can be effectively used after radical trachelectomy or cervical conization.

The strengths of FD-2, the newly developed anticervical stenosis device, are as follows. EVA copolymers, which FD-2 is composed of, are chemically inert, highly stable, nonbiodegradable, transparent, “rubber-like” in softness and flexibility, and are extremely tough, and they are considered to be nontoxic, with no known adverse effects on human health, and noncarcinogenic. As the safety of long-term FD-1 usage (>2 years), an intrauterine contraceptive device, has been demonstrated in women [6], we expect that FD-2 can be left in the uterus for long periods to maintain the dilatation of the cervical ostium. Moreover, this anticervical stenosis device may be more effective than the previously reported IUDs tied with nylon threads [16] because the body of FD-2 facilitates a more constitutive dilation force on the cervix and the drainage of menstrual blood and uterine fluid. Moreover, we believe that the use of FD-2 may have important clinical implications. First, this device eliminates the need for suturing the antistenosis device to the uterine cervix. Second, because of the highly elastic nature of the material, its insertion and removal can be performed safely and easily and the procedure is reproducible. In our experience, insertion and removal take < 10 s. Third, FD-2 can be used after cervical conization to prevent postoperative cervical stenosis. Moreover, during the indwelling period, it can act as an IUD, as the head of FD-2 is identical to the established IUD, FD-1.

This study had limitations. First, the safety of the FD-2 material, EVA copolymers, in an oncological setting remains largely unknown. Second, the clinical aspect of the study is a case series including a very small number of patients (*n* = 3). Although FD-2 met the minimum criteria required for a medical device in bench testing and was employed as an antistenosis device in three cases after obtaining informed consent, for future multi-institutional clinical trials, FD-2 needs to clear additional tests, such as sterile package testing, packaging compatibility testing, or distribution risk assessment. Third, owing to the short follow-up period, long-term safety and efficacy data are not available. Fourth is the duration of FD-2 indwelling in our cases; we indwelled FD-2 for 5–10 weeks based on the previous studies in which the Foley catheter has been used as an antistenosis device for 3 days to 8 weeks after radical trachelectomy [2] or a Japanese study in which intrauterine contraceptive device FD-1 remained indwelled after radical tracherectomy until patients experienced two cycles of regular menstrual periods [3]. However, no evidence-based guidelines exist regarding the duration for which an anticervical stenosis device should be placed in the uterus. To demonstrate the safety and utility of FD-2 in preventing cervical stenosis, as well as to investigate the optimal FD-2 indwelling period, a prospective clinical trial with a large randomized population is required in the future.

## 5. Conclusions

We have developed FD-2, a novel anticervical stenosis device, using EVA copolymers. FD-2 consists of three parts (body, neck, and head) and has met the minimum criteria required for a medical device in bench testing. Our preliminary data may suggest that FD-2 can be safely and easily used in patients who have undergone radical trachelectomy or cervical conization. We hope that the utility of this novel anticervical stenosis device will be further investigated in future clinical studies.

## 6. Patents

FD-2, a newly developed anticervical stenosis device, is applied for an international patent (Application No.22FLC02P).

## Figures and Tables

**Figure 1 bioengineering-10-01032-f001:**
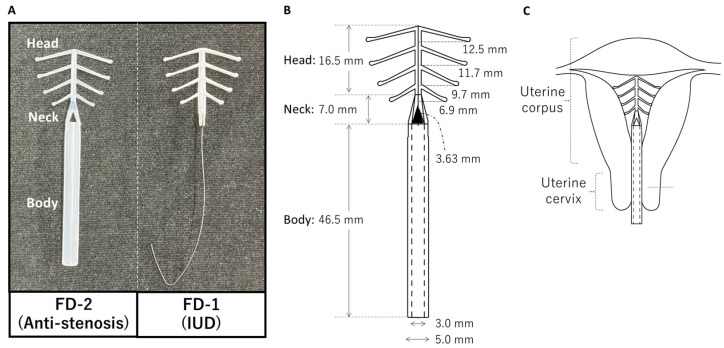
FD-2, a newly developed antistenosis device. (**A**) Left: a picture of FD-2; right: FD-1, an intrauterine contraceptive device (IUD), as a reference. (**B**) blueprint of FD-2. (**C**) Schematic illustrations of FD-2 indwelling.

**Figure 2 bioengineering-10-01032-f002:**
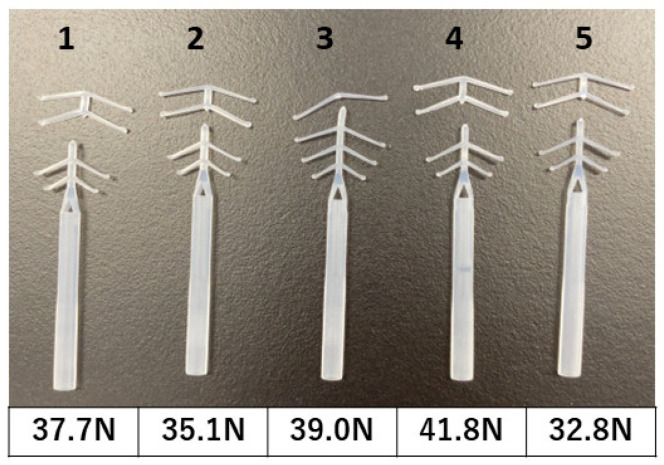
Results from the tensile test.

**Figure 3 bioengineering-10-01032-f003:**
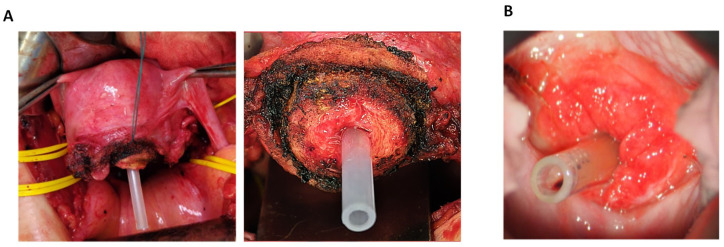
Pictures showing FD-2 used in patients with cervical cancer. (**A**) intraoperative views just after FD-2 insertion, before anastomosis of the uterine cervix (**Left**, distant view; **Right**, magnified view). (**B**) postoperative view (4 weeks after surgery).

**Table 1 bioengineering-10-01032-t001:** Results from the extractables and leachables test.

	Reference Score	Test Value
Transparency test	Transparent	Transparent
pH *	≤1.50	0.7
Foaming test	≤2 min **	30 s
Heavy metals *	≤1.0 ppm	≤1.0 ppm
Potassium permanganate-reducing substances *	≤1.0 mL	0.2 mL
Reside on evaporation *	≤1.0 mg	0.0 mg

* The difference in pH between the Test solution and Reference solution. ** Time needed for the almost complete disappearance of the foam generated.

## Data Availability

The data that support the findings of this study are available from the corresponding author, [S.M.], upon reasonable request.

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
