# Peer review of "FD-2, an Anticervical Stenosis Device for Patients Undergoing Radical Trachelectomy or Cervical Conization"

_bioengineering, 2023, doi:10.3390/bioengineering10091032_

Round 1
Reviewer 1 Report
Article and finding it interesting, however difficult to understand. Please write a technical paper, not a medical report.
The intro is too general; please add more related LR.
Methodology: Suggest to add flow to the study to make it easier to understand.
Explain it step by step.
Result: Overall, it's good.
Conclusion: Too short and not answering the objective of the study.
Suggest to proofread
Author Response
Responses to Reviewer 1
Comment 1:
Article and finding it interesting, however difficult to understand. Please write a technical paper, not a medical report.
Response:
I agree with the reviewer’s comment: the paper should be as technical as possible. However, for the future clinical development of anti-stenosis device FD-2, its preliminary use and the data showing its potential safety in patients are also important. To be more technical, we have omitted Table 2 and changed a title. But please let us include the case series part including Figure 1 in the revised manuscript.
Comment 2:
The intro is too general; please add more related literature reviews.
Response:
As suggested, we have revised an “Introduction” by including some more literatures (lines 49-57 of the revised manuscript).
Comment 3:
Methodology: Suggest to add flow to the study to make it easier to understand. Explain it step by step.
Response:
As suggested, we have added a flow to explain the study step by step (Supplemental Figure 1). We have indicated this in lines 62-63 of the revised manuscript.
Comment 4:
Conclusion: Too short and not answering the objective of the study.
Response:
As suggested, we have revised the conclusion of the study (lines 232-237).

Reviewer 2 Report
Dear Authors,
I read with pleasure your very interesting work “FD-2, a new anti-cervical stenosis device for patients undergo- 2 ing radical trachelectomy: bench testings and a case series”.
I found it very original and of great use in clinical practice. Cervical cancer, unfortunately, although declining in some countries of the world as an incidence thanks to the tremendous work done by regional screening and even earlier with HPV vaccination, still afflicts too many young female populations. These women, often in their 30s, have not yet exhausted their desire for offspring and, if possible, would be eager to receive fertility sparing treatment. The surgical procedures that can be performed in these cases, and depending on the clinical and histological stage of the disease, however, provides for gradually increasing radicality at the expense of parameters that inevitably impact the outcomes of any future pregnancies. In addition, trachelectomy resection, which is different depending on whether it is vaginally, laparoscopically, or laparotomically, carries a significant risk of cervical stenosis with its attendant complications.
Studies in the literature are currently inconclusive on what strategies to adopt in the prevention and/or treatment of this condition. The currently known devices (IUD, smit sleeve, foley catheter) to the extent that they have been used, have not yielded significant results for standardized large-scale use. Therefore, your work fits into a context of undeniable need and innovation.
As you have already anticipated, these are initial data and will require much further work to prove their effectiveness.
In any case, at the current state of the evidence you listed, I ask:
- Is to be implemented the introduction section
- the material of the device, which has already been used for FD1s (contraceptive devices), are there any studies demonstrating their harmlessness in an oncological setting such as that of the patients who would be receiving it? at line 194, you mention this issue but no literature notes are reported. Therefore, I would ask you to add them and better justify what these evidences are based on.
- it is clear that leaving the device in place while waiting for a menstrual cycle is intended to assess non-interference with normal blood flow, but, I wonder what the rationale was for leaving it in place for only 2 cycles.
- lines 89-90: "previously reported" Where have the authors previously reported it?
- the properties tested and illustrated in the table, are they complete with everything that should be tested of the device material? could you add a bit of detail on this.
- Have you planned how to proceed in future studies?
in the discussion, mention is made of the greater ease of insertion and removal and of the non-need for cervical suture for device placement: what is the rationale? is it due to the greater hardness as well as elasticity of the device?
Dear Editor,
the quality of English is appropiate, requiring little correction.
Author Response
Responses to Reviewer 2
Comment 1:
The material of the device, which has already been used for FD1s (contraceptive devices), are there any studies demonstrating their harmlessness in an oncological setting such as that of the patients who would be receiving it? at line 194, you mention this issue but no literature notes are reported. Therefore, I would ask you to add them and better justify what these evidences are based on.
Response:
Thank you for the reviewer’s thoughtful comment. As mentioned in lines 185-189, in a relatively large study including 65 cervical cancer patients who underwent radical tracherectomy, FD-1 were routinely utilized to prevent intracavity adhesion [Arch Gynecol Obstet. 2010,281,887-9.]. However, the safety of FD-1 in an oncological setting has not been fully investigated and remains unknown. We have indicated this as a limitation of the current study (lines 190-191 and 214-215 of the revised manuscript).
Comment 2:
It is clear that leaving the device in place while waiting for a menstrual cycle is intended to assess non-interference with normal blood flow, but, I wonder what the rationale was for leaving it in place for only 2 cycles.
Response:
As described in line 148, we indwelled FD-2 for 2 menstrual cycles (5-10 weeks) based on the previous studies in which the Foley catheter has been used as an anti-stenosis device for 3 days to 8 weeks after radical trachelectomy (Eur J Cancer. 2015,51,1751-9.) or a Japanese study in which intrauterine contraceptive device FD-1 was remained indwelled after radical tracherectomy until patients experienced two cycles of regular menstrual periods (Gynecol Oncol. 2009,115,51-55.). However, as the reviewer pointed, no evidence-based guidelines exist regarding the duration for which an anti-cervical stenosis device should be placed in the uterus. So, to demonstrate the safety and utility of FD-2 in preventing cervical stenosis as well as to investigate the optimal FD-2 indwelling period, a prospective clinical trial with a large randomized population is required in the future. We have described this in lines 221-230 of the revised manuscript.
Comment 3:
Lines 89-90: "previously reported" Where have the authors previously reported it?
Response:
We did not report the current cases in other journals. In lines 89-90 of the original manuscript, we wanted to say “We performed radical trachelectomy via vaginal or retroperitoneal approach using the same procedures that were reported previously [9,10]”. We have revised the wordings. As reproperitoneal radical tracherectomy had been developed by the first author (S.M.) of this article [9,10], and has been performed only in limited institutions, citations are required. To avoid confusion, we have revised a wordings (lines 101-102 of the revised manuscript).
Comment 4:
The properties tested and illustrated in the table, are they complete with everything that should be tested of the device material? could you add a bit of detail on this.
Response:
Bench testings illustrated in the manuscript (Tensile test, extractables and leachable test) are not all that should be tested for the device material. To conduct a clinical study or to be approved, we need additional tests: i.e. sterile package testing, packaging compatibility testing. We have indicated these in the revised version (lines 216-220).
Comment 5:
Have you planned how to proceed in future studies?
Response:
We are currently consulting the Pharmaceuticals and Medical Devices Agency in Japan on how to proceed the future clinical studies. We have indicated this in the revised version (lines 65-67 and Supplemental Figure 1).
Comment 6:
In the discussion, mention is made of the greater ease of insertion and removal and of the non-need for cervical suture for device placement: what is the rationale? is it due to the greater hardness as well as elasticity of the device?
Response:
Thank you for the reviewer’s comment. Because the pliable arms in the head part of FD-2 can be bended back and forth, FD-2 insertion into the uterine cavity ought to be easy by holding the body of FD-2 and pushing its head into the uterine cavity through the cervical canal. FD-2 removal ought to be easy as well by gently pulling the body using forceps. The pliability of these arms can be achieved by the highly elastic nature of EVA copolymers. In the absence of force to the FD-2, a physiological condition, the arms do not bend and prevent FD-2 from falling out and help maintain the body part in the proper position within the cervical canal. We have indicated these in the revised version (lines 72-79).
